# Real-time feedback control of the impurity emission front in tokamak divertor plasmas

T. Ravensbergen [1,2✉], M. van Berkel [1✉], A. Perek [1,3], C. Galperti[3], B. P. Duval[3], O. Février [3], R. J. R. van Kampen [1,2], F. Felici [3], J. T. Lammers [2], C. Theiler [3], J. Schoukens [4,5], B. Linehan[6], M. Komm[7], S. Henderson [8], D. Brida[9] & M. R. de Baar[1,2]

In magnetic confinement thermonuclear fusion the exhaust of heat and particles from the core remains a major challenge. Heat and particles leaving the core are transported via open magnetic field lines to a region of the reactor wall, called the divertor. Unabated, the heat and particle fluxes may become intolerable and damage the divertor. Controlled 'plasma detachment', a regime characterized by both a large reduction in plasma pressure and temperature at the divertor target, is required to reduce fluxes onto the divertor. Here we report a systematic approach towards achieving this critical need through feedback control of impurity emission front locations and its experimental demonstration. Our approach comprises a combination of real-time plasma diagnostic utilization, dynamic characterization of the plasma in proximity to the divertor, and efficient, reliable offline feedback controller design.

[1] DIFFER - Dutch Institute for Fundamental Energy Research, Eindhoven, Netherlands. [2] Department of Mechanical Engineering, Control Systems Technology Group, Eindhoven University of Technology, Eindhoven, Netherlands. [3] École Polytechnique Fédérale de Lausanne (EPFL), Swiss Plasma Center (SPC), Lausanne, Switzerland. [4] Department ELEC, Vrije Universiteit Brussel, Brussels, Belgium. [5] Department of Electrical Engineering, Control Systems Group, Eindhoven University of Technology, Eindhoven, Netherlands. [6] Plasma Science and Fusion Center, Massachusetts Institute of Technology, Cambridge, MA, USA. [7] Institute of Plasma Physics of the CAS, Prague 8, Czech Republic. [8] CCFE, Culham Science Centre, Abingdon, Oxon, United Kingdom. [9] Max-Planck-Institut für Plasmaphysik, Garching bei München, Germany. ✉email: Timo.Ravensbergen@iter.org; M.vanBerkel@differ.nl

Nuclear fusion is one of the few future energy sources which has the potential to fulfill our energy demands whilst remaining virtually inexhaustible, inherently safe, and deployable in densely populated regions. One of the major unresolved challenges in nuclear fusion power research is the heat and particle exhaust[1–3].

This paper reports on experiments on the most advanced fusion plasma device configuration, the tokamak (see Fig. 1). The magnetic field lines in a tokamak, that charged particles will follow as they precess, self-arrange into what are effectively toroidally symmetric, nested surfaces. To diminish interaction with the vessel wall, a poloidal field null, termed X-point, is introduced using shaping coils. The plasma is thus divided into a confined region where the magnetic field lines close upon themselves, and an outside, unconfined, region separated by the magnetic surface that goes through the X-point, termed the separatrix. The two branches of the X-point that do not enclose the confined region eventually lead to the machine walls and are called the divertor legs and their impact points, the divertor strike points. The power reaching these targets, if unmitigated, will exceed material limits for projected fusion reactor conditions such as ITER[4,5] and DEMO[1].

Reducing the power density in the plasma exhaust from the confined plasma boundary (termed upstream) to the targets (termed downstream) is being explored by adding radiating and/or collisional loss processes along the divertor legs. These processes, when present to a sufficient degree, lead to the formation of a pressure and power drop that is not observed during usual operation and is termed detachment. The power decrease is induced by injection of hydrogenic and impurity gas and most often accompanied by a strong temperature decrease along the divertor leg. Such a decrease can be observed as a strong radiation intensity decrease[6], in certain impurity species' radiation when the electron temperature becomes insufficient to excite that impurity's atomic states. In general, if this radiation front enters the confined plasma, it can degrade the confinement[7]. On the other hand, when the divertor detachment is insufficient, the

divertor targets' heat load becomes excessive[8]. This simplified description is complicated in practice by plasma and/or impurity transport, plasma instabilities (ELMs,[9, p. 409], sawtooth instabilities[9, p. 365]), and plasma flows in the divertor region that these divertor legs occupy. Therefore, to support the DEMO divertor design, existing facilities are being upgraded[10], and a new tokamak facility to study the plasma exhaust is under construction[11,12].

In view of changes in plasma behavior, feedback control of the divertor plasma conditions is considered vital to assure safe, high-performance, operating conditions in future fusion devices[6]. Various feedback control solutions involving detachment have been tested experimentally, using controlled variables based on either local target measurements or from spatially resolved diagnostics. Implementations based on target measurements rely on, for example, tile current[13] or target ion saturation current measurements[14], or target proximity thermocouples[15]. Among the spatially resolved methods are those based on Thomson scattering[16] and radiated power from bolometry[17,18] or AXUV[19,20]. These experiments rely on (impurity) gas fueling through controlled valves as actuators. In addition, supersonic molecular beam injection was successfully applied as feedback actuator[21].

In this paper we demonstrate a strategy to control impurity emission front locations, as a proxy for divertor detachment, on the Tokamak Configuration Variable (TCV)[22]. (i) We apply real-time analysis of multi-spectral video images obtained from the MANTIS diagnostic[23] to reconstruct the poloidal location of a chosen spectral line's emission front. (ii) We perform *system identification* experiments, both in low (L) and high (H) confinement modes[24], to characterize the dynamic relationship between gas valve actuation and displacement of the emission front when above the divertor target. (iii) We perform an offline feedback design using these characterizations, leading to dedicated feedback controllers for both confinement modes. (iv) We verify experimentally that the controllers function successfully during their first closed-loop feedback application. These dedicated experiments request different reference trajectories of the emission front location, where the active gas fueling changes are used to track the requested waveforms.

## Results

**Real-time reconstruction of the C-III emission front.** The state of divertor detachment is generally well characterized by the divertor electron temperature[25]. In this work, we control the position of the C-III emission front (465 nm) along the divertor leg. This emission front has been shown to be a good marker for a relatively low temperature region[26,27]. Note that the divertor plasma is mostly tied to the magnetic field lines, such that the front movement is predominantly in the direction along the divertor legs and approximately toroidally symmetric. This reduces and simplifies the control of the emission front position to a single dimension: the distance along the divertor leg to the target, denoted with $L_{pol}$ in Fig. 1 and throughout this paper.

*Experimental setup.* A poloidal cross-section of the TCV tokamak is shown in Fig. 2a, together with a typical diverted plasma equilibrium. TCV is a medium-sized tokamak whose advanced distributed control system[28] facilitates rapid implementation of various control strategies. Optional physical *baffling* in the divertor increases neutral particle compression, facilitating detachment[26]. Such baffles were installed during a majority of the experiments presented here. The multi-spectral imaging diagnostic MANTIS installed on TCV acquires up to ten spectrally filtered 2D images of an identical view of the divertor plasma[23].

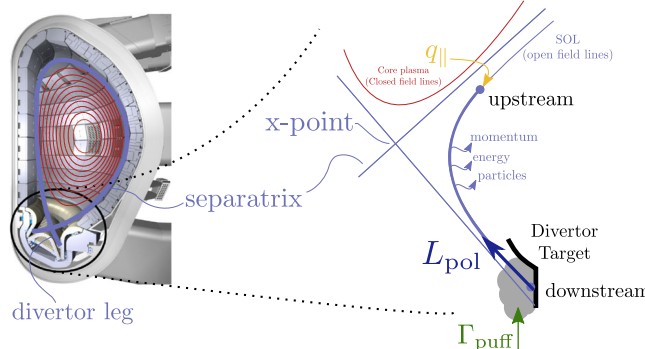

**Fig. 1 Detachment in a tokamak divertor plasma.** Left: Diverted plasma configuration in a tokamak with schematic poloidal flux surfaces. The confined region (red curves) is separated from the open field line or scrape-off-layer (SOL) region by the separatrix, shown as a thick blue line. Right: Summary of the relevant aspects of the detachment process. High heat fluxes densities $q_{\parallel}$ along the SOL field lines are mitigated by a gas buffer, which is controlled by local gas puffing $\Gamma_{puff}$. This causes a combination of volumetric loss processes (e.g., momentum, energy, and particle losses), resulting in significantly lower downstream target pressures and temperatures compared to the upstream conditions. These conditions give rise to the formation of emission fronts. The distance along the divertor leg to the target of the impurity emission front (the arc length) is denoted by $L_{pol}$. (Cross-sectional mock-up reproduced from: (c) ITER Organization, http://www.iter.org/).

The system is mounted on an outside lower (divertor) port, whose view is shown in Fig. 2b. In this image, the carbon plasma facing components on the TCV floor and inner wall are clearly visible, but the field-of-view is restricted from the top by the outer baffle.

*Real-time tracking of the emission front.* Figure 3 shows raw C-III spectrally filtered images for various snapshots during a core plasma particle density ramp. Depending on the electron temperature of the divertor plasma, emission can be present across the entire divertor leg (shown in Fig. 3a), can cease along the divertor leg where the plasma temperature is sufficiently low (see Fig. 3b), or enter the confined plasma region (shown in Fig. 3c).

By looking for sharp gradients, the detection algorithm[29] first locates the divertor leg in these spectrally filtered images. It then tracks the location of the 50th percentile (decrease) of light emission along the divertor leg: a common definition of the emission front that is extensively discussed in[30,31] together with its physical interpretation. Strictly speaking, this emission decrease along the leg is defined for tomographically

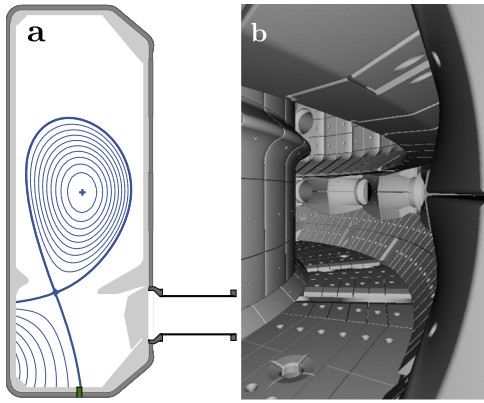

**Fig. 2 The TCV tokamak and MANTIS view. a** Cross-section of TCV showing the plasma magnetic flux surfaces (blue) for diverted discharge #65349, the vacuum vessel including baffles (light gray), the MANTIS view port, and the location of the fueling gas valve (green). **b** Rendering of the MANTIS view into the TCV vacuum vessel[52]. The field-of-view is restricted by the TCV floor (bottom) and outer baffle structure (top).

reconstructed images, where the line collected integrated light intensity is inverted to obtain local emissivities[32,33]. This procedure corrects for changes in intersection length between each line-of-sight and the plasma, but is computationally too expensive for real-time use. Fortunately, taking the raw image as input (which thus contains toroidally integrated light along the line-of-sight) yields a well resolved and sufficiently parametric measurement of the emission front position[29]. Figure 3 shows the leg and emission front detection in the divertor plasma. A subsequent mapping of the pixels associated with the divertor leg and emission front to the poloidal coordinates of TCV gives the parameter of interest $L_{pol}$ (Fig. 1). Figure 3d shows this mapping in blue for the divertor leg in Fig. 3a, with the full tomographic inversion of the same frame in the background. The details of the detection algorithm, including its accuracy and comparison with post-discharge analysis are highlighted in ref. [29].

**Dynamic characterization of the C-III emission front in TCV.** In the following, we discuss the experiments used to identify the dynamics of the displacement of the C-III emission front in the poloidal cross-section of TCV as a result of variations in the fueling. Reliable, systematic offline feedback controller design requires an understanding of the dynamics of the to-be-controlled system. This is quantified by a *frequency response function* (FRF) that can be obtained from dynamic modeling[34], or directly from experimental data[35]. The resulting FRF can be used for offline assessment of performance limitations and stability margins[34]. Accounting for the system dynamics within the physical modelling of divertor plasmas is challenging, as the dynamics themselves are often strongly affected by kinetic effects in the plasma-neutral interaction calling for Monte Carlo methods[1,4,36,37]. We instead apply techniques from system identification to extract sufficient information directly from experiments. Such an approach leads to a locally valid dynamic description around an operating point that can be used in subsequent controller design. In the *loopshaping* method applied here, the controller design consists of defining and assessing stability and performance criteria in the frequency domain for different controller choices and their parameters.

**System identification experiment design and goal.** Figure 1 and Fig. 4 schematically show the to-be-identified dynamical system under consideration. The gas puff $\Gamma_{puff}$ acts as a dynamic,

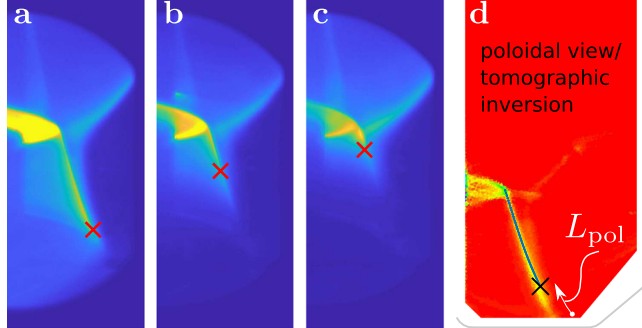

**Fig. 3 Progressive cooling of the TCV divertor.** C-III emission in TCV tokamak discharge #62154 for increasing radiative cooling of the divertor. The detected emission front is indicated in red. **a** Emission throughout the divertor leg: Emission front touches the wall. **b** The emission front creeps up toward X-point (**c**) Emission front close to the X-point. **d** Poloidal mapping of the divertor leg (blue) and emission front (black cross) in (**a**) in the TCV vacuum vessel with plasma emissivity from tomographic reconstruction in the background. The control parameter $L_{pol}$, the arc length from emission front to divertor target, is highlighted in white.

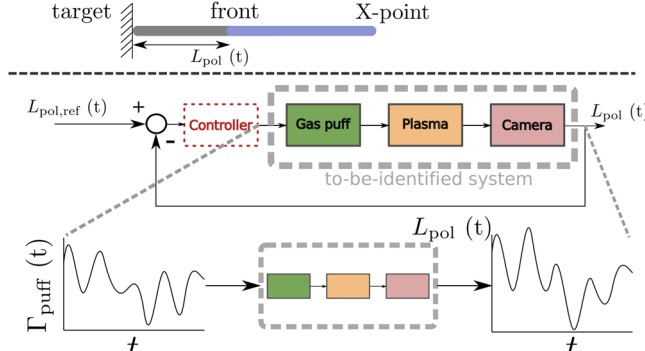

**Fig. 4 Schematic representation of system identification and feedback controller.** A simplified, one-dimensional cartoon of the dynamic SOL plasma (Fig. 1) under consideration. The dynamic response of $L_{pol}$ upon fueling needs to be characterized for feedback controller design. Even though many parameters affect the divertor plasma, only the input-output behavior in the gray box has to be captured by a transfer function model. This transfer function model then describes the combined linearized dynamics of the gas valve, plasma, and camera system.

time-varying input to the SOL plasma that leads to an emission front position $L_{pol}$ (i.e., the emission front is no longer located at the divertor target where $L_{pol} = 0$) which is used as output of the system. The piezo-type gas valve[38] that controls $\Gamma_{puff}$ has its own internal feedback controller aiming to track a gas puff request. We directly control the voltage on the piezo valve: Higher voltage increases the gas flow, as the valve opens further. Besides $\Gamma_{puff}$ several external parameters may intervene and affect $L_{pol}$. Particle removal in the divertor is dominated by absorption by the first wall material, known as wall pumping, which cannot be actively controlled. The variations in the heat flux from the core plasma $q_{\parallel}$ are considered to be another unknown disturbance. Significant additional disturbances on $L_{pol}$ are present due to coupling with the vertical position of the plasma and, depending on the plasma conditions, instabilities like ELMs and sawteeth, which can temporarily disturb $L_{pol}$ by changing the carbon emission profile.

The dynamic multi-dimensional relationship between $\Gamma_{puff}$ and $L_{pol}$ is simplified in a one-dimensional control problem, which is expressed in abstract form in Fig. 4. Modifying $\Gamma_{puff}$ results in a strongly correlated response of $L_{pol}$, which needs to be captured in an FRF. This is shown bounded by the gray box in Fig. 4 and describes the combined dynamics of gas valve, plasma, and camera.

System identification provides a framework to efficiently sample frequency points in a system's FRF. This measurement allows direct controller design via loopshaping, and verification of the accuracy and applicability of existing dynamic *transfer function* models[35]. In principle both an FRF measurement or a transfer function model can be used for offline controller design with loopshaping. However, in this particular case, the transfer function model is preferred as the FRF measurement consists of a limited number of frequency data points. Therefore, a simple transfer function model is fitted through the measured FRF points, which is then used for subsequent controller design. After controller tuning and assessment, the controller (in red in Fig. 4) connects the same two channels $L_{pol}$ and $\Gamma_{puff}$ previously used for system identification in a closed-loop.

**Choice of gas valve excitation signal**. A dynamical system is often characterized by applying a perturbation to its input. Applying a kick or white noise perturbation allows an identification of the entire FRF for all frequencies[35]. The applied energy for these perturbations is distributed, however, over the whole spectrum, often leading to poor signal-to-noise ratio (SNR) performance in the output that comprises the FRF. This especially holds for systems with significant external disturbances on the measurement, such that the applied perturbation becomes indistinguishable from the experimental fluctuations. While these disturbances on $L_{pol}$ are generally significantly faster than e.g., the MANTIS measurement or the gas valve, they worsen the FRF content in the lower frequency range of interest due to effects like ghosting and/or aliasing[39]. Although the SNR would be improved by longer measurement times, a typical plasma discharge on TCV lasts 1.5 s. As appropriate measurement time is therefore both expensive and short, and significant disturbance on $L_{pol}$ is present, it becomes more appropriate to maximize the input energy density of the perturbation signal in the relevant frequency range to maximize the SNR. Therefore, a multisine signal on the input side was chosen that only excites specific frequencies[40]. The choice for such a signal leads to higher a priori design effort and less frequency points in the final FRF, but is strongly justified by the improved SNR.

**Experimental plasma scenarios**. Two sets of experiments were performed: An L-mode scenario, and an H-mode scenario, which

employed 850 kW of additional neutral beam injection. The divertor plasma dynamics are known to differ between L-mode and H-mode[36], calling for separate system identification for each case. A significant high frequency disturbance on $L_{pol}$ can result from ELM activity in the H-mode scenario. The ELM size progressively decreases for the scenario used here, ending in the attractive 'small-ELM' regime, where the ELM strength is relatively small[41]. During all experiments a core density feedback controller was used together with a programmed density ramp to bring the divertor plasma to a sufficiently cooled state. The output of this controller is then frozen and the perturbation signal started. This perturbation (schematically shown in Fig. 4) is summed with a feedforward waveform that keeps the core density approximately constant. The SNR was maximized by employing an optimized perturbation, whose design is discussed in the Methods.

**Perturbative measurements and transfer function estimation**. Figure 5 shows the results of the perturbation experiments in both the time and, by applying the Discrete-Fourier Transform (DFT), the frequency domain for all considered cases. The response of the divertor plasma in terms of $L_{pol}$ is shown in red, and the gas valve perturbation signal in black. In the experiment, this zero-mean perturbation is summed with a feedforward gas trace that keeps the core density approximately constant. The total signal that is sent to the gas valve in these experiments is therefore always positive. For all studied cases, the response of the plasma is dominantly linear, as the largest response on the output side is at frequencies that match the input[35] in the DFT. However, as discussed, significant uncertainty is present on the signal due to vertical plasma displacement and sawteeth (L-mode) with additional contribution from ELM disturbances for the H-mode case. The magnitude of the total noise, including originating from less well-known sources, is estimated in the DFT by averaging the response for frequencies higher than 1/4 of the sampling frequency. A black dashed line shows this noise floor estimate. During the unbaffled L-mode experiment, Fig. 5a, MANTIS was operating on 200 Hz instead of 800 Hz, contributing to a higher noise level compared to the baffled L-mode.

A comparison between experiments in terms of the DFT leads to the observation that nonlinear contributions of ELMs to the system dynamics result in significantly higher responses at non-excited frequencies[42] compared with the ELM-free L-mode case. These responses are clearly above the indicated noise floor, but well below the response at excited frequencies. In all experiments, a decrease in plasma response amplitude at higher frequencies (known as roll-off) is apparent. This indicates a system that is significantly slower than the typical time-scales of plasma processes. Combined with the significant overall system delay, this limits the bandwidth a possible feedback controller can achieve[34].

From these identification experiments two important conclusions can be drawn: (1) The disturbance on the measurement due to vertical plasma oscillation, sawteeth and/or ELMs, and other sources is significant. However, as we applied suitable multi-sines as perturbation signals, we could raise the responses at the applied frequencies well above the noise floor in the DFT. (2) Especially for the H-mode experiment in **c**, $L_{pol}$ is significantly drifting upwards in time. Further, a *transient effect* in the early phase of the measurement is apparent in the DFT, leading to a high response at frequencies below the lowest excited frequency[35]. Both effects need to be accounted for in subsequent data processing to obtain the FRF. Here, we apply the Local Polynomial Method (LPM) to estimate the transient and drift effects in the data[42]. Then, the FRF is obtained by taking the ratio

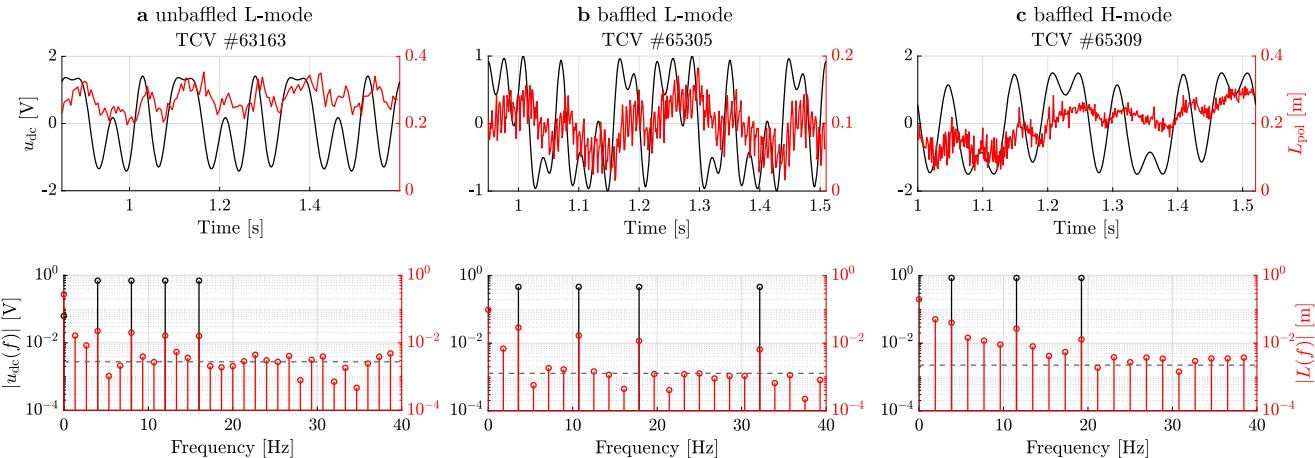

**Fig. 5 System identification experiments.** Overview of system identification results as obtained on the TCV tokamak. Each separate measurement plot contains as a top trace the time evolution of the multi-sine perturbation (black) and response in the emission front $L_{pol}$ (red). In the experiment, the perturbation is summed with a feedfoward waveform to yield the total (always positive) gas valve command. Each bottom plot shows the Discrete-Fourier transform (DFT) of the perturbation (black) and the plasma response (red), a dashed black line shows an estimate of the noise floor. This noise floor in the unbaffled L-mode experiment (**a**) is high due to the lower (200 Hz) MANTIS frame rate. The baffled L-mode experiment (**b**) has the highest SNR with MANTIS running at 800 Hz. In contrast, the H-mode experiment (in **c**) shows a high measurement noise floor due to the additional disturbance in the form of ELMs, which also cause significant contributions at non-excited frequencies. Furthermore, the plasma response is overall evidently smaller compared to the L-mode cases in (**a** and **b**). Finally, the less favorable experimental conditions in H-mode lead to a more pronounced drift of $L_{pol}$ over time in (**c**), which is compensated in further analysis.

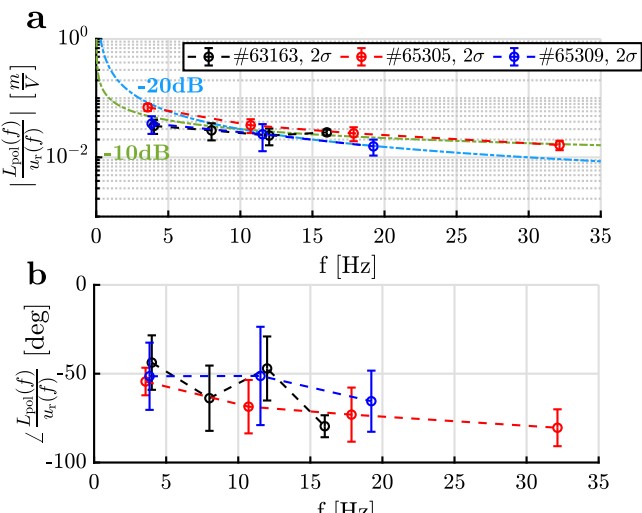

**Fig. 6 Overview of estimated frequency responses.** Magnitude (plot **a**) and phase (plot **b**) of the FRFs of all perturbation studies as obtained with the LPM[42]: Unbaffled L-mode #63163 baffled L-mode #65305, and baffled H-mode #65309. The gain of the H-mode experiment is lower and close to the unbaffled L-mode case. The phase characteristic of all FRFs is reminiscent of an irrational transfer function model[44]. $2\sigma$-Error bars are calculated from the LPM[42].

of the DFT of $L_{pol}(f)$ and $\Gamma_{puff}(f)$ with the estimated transient and drift effects removed.

Figure 6 shows the obtained FRFs for all experiments, as obtained with the LPM. The differences in dynamic response between experiments are minor, and appear to be mostly in the steady-state gain of the FRF. One aspect of the frequency response characteristic is reflected in the phase of the FRF, i.e., the phase difference between $L_{pol}(t)$ and $u(t)$, Fig. 6b. We observe that the dynamics we measure are considerably different from that expected from standard gas-valve dynamic models alone. Such gas valve models are often in rational form[43] and have a low, finite number of poles

and zeros in their transfer function model. For such rational systems, provided they are linear and minimum-phase[34], each pole results in a $-20$ dB/decade gain change and $-90°$ phase change. Therefore, a first order system (with one pole) has a slope of 0 for frequencies below its pole, whereas for higher frequencies the slope is $-20$ dB/decade (accompanied by a phase of $-90°$). However, we measure (within the frequency window) an approximately constant magnitude slope of $-10$ dB/decade (Fig. 6a), with gradual phase decrease starting from ~45° at our lowest excited frequency (Fig. 6-bottom). This indicates an irrational transfer function model, originating from a partial differential equation[44,45]. Although this may merit further investigation, more identification experiments are needed to be conclusive. We therefore focus on robustness and stability over performance in the proof-of-principle control design by fitting a simple gain-delay transfer function model through the obtained FRFs. This transfer function model is of the form:

$$S_{\Gamma_{puff}\to L_{pol}}(j\omega) = Ke^{-\theta j\omega}, \qquad (1)$$

where $K$ denotes the open-loop gain, and $\theta$ the total time delay of the system. The frequency $\omega$ is in radians/second. As such, the delay $\theta$ incorporates the total delay from gas valve signal to actual front position in a simplified description, allowing subsequent feedback controller design with sufficient stability margins[34]. This controller design and its application are discussed next.

**Feedback control experiments.** In the following, we will show the achieved feedback control results of the C-III emission front for two plasma discharges. With all L-mode and H-mode transfer function models identified, the loopshaping method was used to tune three proportional-integral (PI) controllers on Eq. (1) for the various configurations in Fig. 5. The PI controller has the following FRF:

$$C = K_p \frac{j\omega + K_i}{j\omega}, \qquad (2)$$

in which $K_p$ denotes the controller gain, and $K_i$ the zero (cut-off frequency) of the integral action. The controllers were designed to

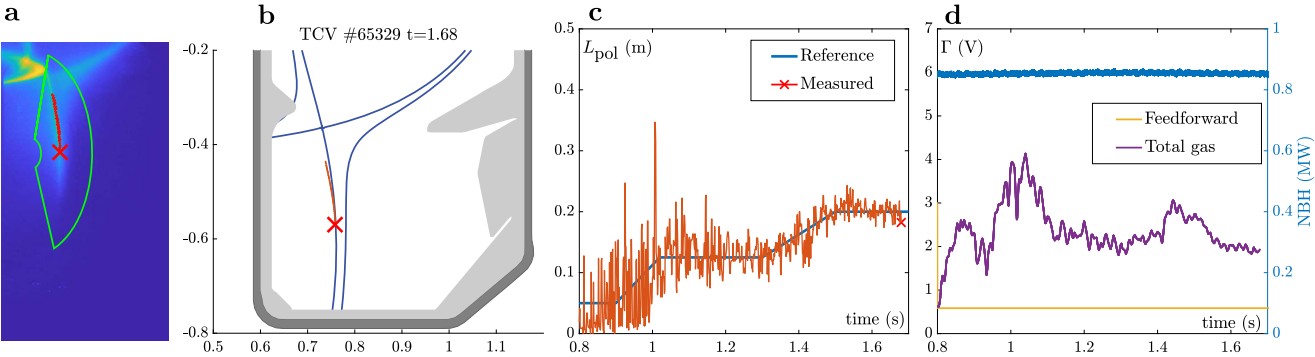

**Fig. 7 H-mode feedback control results.** Feedback control of the C-III emission front in H-mode confinement for a stair-case like reference trajectory. **a** Snapshot of a MANTIS image with region-of-interest, detected emission front, and divertor leg. **b** Mapping of the detected divertor leg and front to the poloidal cross section of TCV, together with the (offline) LIUQE magnetic equilibrium reconstruction of the separatrix[46]. **c** Reference tracking with the requested $L_{pol}$ in blue, and real-time reconstruction in orange. The controller is active from $t = 0.8$ s onward. The ELM size and frequency change during the discharge due to the increase in core density, leading to a progressively smaller disturbance on the emission front measurement. **d** The gas valve feedfoward waveform in yellow, with superimposed feedback control signal in purple. The delivered NBH power is plotted in blue.

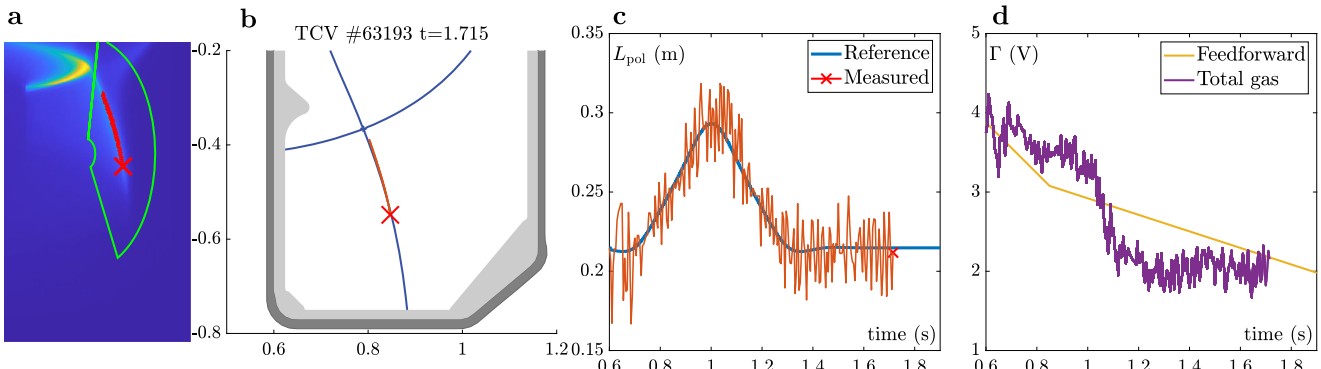

**Fig. 8 L-mode feedback control results.** Feedback control of the C-III emission front for bump-like reference trajectory in L-mode. **a** Snapshot of MANTIS image with ROI and detected emission front. **b** Detected divertor leg and emission front in TCV cross-section (red) and magnetic equilibrium reconstruction (blue). **c** Time trace of the C-III emission front (red) and its reference (blue). These experiments were performed with MANTIS running on a lower 200 Hz, leading to a lower SNR. The feedback controller is active from $t = 0.6$ s onward. **d** Gas feedforward (yellow) and total feedback command (purple). The latter shows how initially the gas flux is increased to move the front upwards, followed by a decrease of gas to move the front back down.

achieve a closed-loop bandwidth of 7 Hz. This choice of bandwidth allows sufficient performance for clear reference tracking of $L_{pol}$ during the experiment, and simultaneously, sufficient robustness in terms of phase margin[34]. To achieve the same bandwidth for both confinement regimes, the controller gains were adjusted to compensate for the differences in the identified system gain $K$ and delay $\theta$ in (1). The additional integral action in the controller increases tracking performance in the low-frequency range, but has a zero to reduce the lowering of the phase margin for frequencies around the bandwidth. For the H-mode experiment, a low-pass filter (with its passband edge frequency at 40 Hz) was added to reduce the influence of high frequency ELM disturbances on the tracking error.

Like in the identification experiments, in the feedback control experiments the output of the core density feedback controller is frozen when the plasma is significantly cooled (i.e., $L_{pol}$ is between the strike point and the target). The total gas valve command is therefore the sum of the pre-programmed feedforward waveform and the feedback control command using the MANTIS-computed $L_{pol}$. Various reference tracking experiments were performed in both confinement modes. The controller performed as expected upon its first implementation, following its design from data obtained in the same scenario and frequency range.

Figures 7 and 8 summarize the results of two of these experiments. An H-mode discharge with a stair-case reference, and an L-mode experiment with bump-like reference. From left to right, we show the following: **a** A time slice of the raw C-III filtered MANTIS frame near the end of the discharge, with the region-of-interest used for detection, the detected divertor leg, and the emission front indicated. **b** The mapping of the divertor leg to the TCV cross section, and the comparison to (offline) magnetic equilibrium reconstruction[46]. **c** The requested trajectory of $L_{pol}$ (blue), and the MANTIS reconstruction (orange). Finally, in **d** the feedforward waveform of the gas valve (yellow), and the total gas command (feedforward and feedback) is shown in purple.

In both experiments, the feedback controller clearly tracks the requested emission front position by strongly modifying the pre-programmed gas flux. Nevertheless, as for the identification experiments, considerable fluctuations in the emission front position are present, originating from sources that are mostly outside the algorithm itself (see ref. [29] for details). For the L-mode experiments, these originate from vertical position fluctuations of the plasma, with further contributions from the sawtooth activity. The sawtooth frequency is higher than the 200 Hz MANTIS frame rate for this unbaffled experiment,

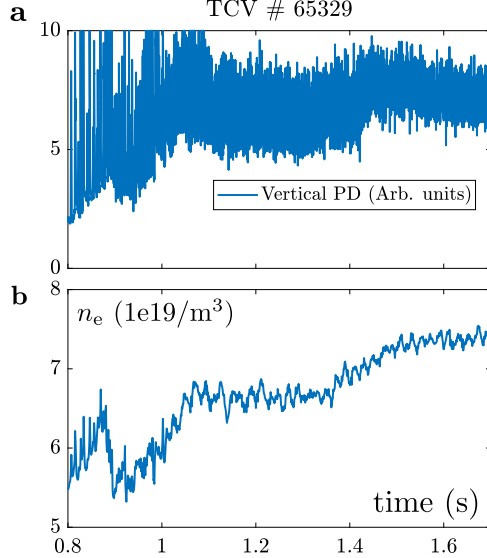

**Fig. 9 Edge Localized Mode evolution. a** Time trace of the vertical photodiode (PD, equipped with with $D_\alpha$-filter) for TCV discharge #65329. The varying ELM size over the course of the discharge is reflected in the amplitude of the photodiode signal. **b** Evolution of the line averaged core density $n_e$ as measured by interferometry.

so the SNR worsens due to aliasing and ghosting effects. For H-mode cases, running at 800 Hz, significant disturbances arises from ELM activity. During the experiment ELMs of various sizes occurred, which is clearly reflected in the size of $L_{pol}$ disturbance. The changing ELM size is further highlighted in Fig. 9 which shows the time trace of the vertical photodiode (PD) signal and the plasma line averaged core density $n_e$. The ELM size, reflected by the height of the peaks in the PD signal, significantly decreases during the discharge and is related to the increase of the plasma density.

## Discussion

In this paper we have identified the dynamic relationship between the gas valve command and the location of the C-III emission front during tokamak operation. The approach of using tailored perturbation signals in a system identification procedure has enabled reliable estimation of the local dynamics with little experimental time. These estimates then allowed offline feedback controller design. By applying the resulting feedback controllers in real-time, a requested reference position for the emission front could be successfully tracked in both L- and H-mode, demonstrating control of a cool divertor state.

This paper reports on the real-time use of a spectral imaging diagnostic to infer the divertor plasma state. The use such a diagnostic for control purposes has some advantages over existing methods. Firstly, our approach yields a 2D spatial measurement of the temperature in the divertor, which is compatible with other heat load mitigation techniques like strike-point sweeping[47]. In addition, a full 2D view is relevant as it has become clear that the onset of detachment is driven from a high density, low temperature region in the private flux region[48]. This makes analysis of diagnostics with a small measurement volume (e.g., Thomson scattering) or with a line-of-sight across the divertor leg (like bolometry or AXUV) challenging. Secondly, MANTIS' line-of-sight is tangential to the plasma, such that (real-time approximations of) inversions are not necessary for control purposes. As such, sampling frequencies up to 800 Hz are achieved, which is

significantly faster than e.g., Thomson scattering (~50 Hz[49]), but somewhat slower than AXUV PDs at 1 kHz[19]. Thirdly, MANTIS enables control in attached, marginally/partially detached, and (strongly) detached conditions, which is not the case for target probes (only attached or marginally detached), or AXUV PDs (only detached)[19,20].

Carbon, used as a first wall material on TCV and the source of C-III emission in the experiments presented here, is not a suitable first-wall material or seeding species for future nuclear devices[50]. This proof-of-principle work will therefore be extended to the control of emission fronts from other radiating species. Metal-walled machines will require some kind of impurity seeding to detach, whereas the presented feedback loop in this work has, to date, only employed the plasma main ion species where there is a strong link between the plasma core density and the radiation front position. The capabilities of the MANTIS diagnostic and the methodologies presented here are well suited to allow tailoring of the radiation distribution in the SOL with multiple impurity lines and/or species. Actually, a system with the same properties as MANTIS on TCV is already under construction for the MAST-U tokamak[51], to gain new insights in the detachment dynamics and achievable controller performance for significantly different divertor designs.

## Methods

**Perturbation signal design**. In our system identification experiments we use a tailored excitation signal that aims to maximize the SNR. Our choice for such a perturbation signal is a multisine waveform. This multisine is characterized by a number of discrete excited frequencies within a bandwidth, and their precise frequencies, amplitudes, and phases. We briefly discuss the method to design these parameters for our specific experiments with the procedure and mathematical background described in more detail in ref. [35].

*Frequency range*. The range of possible frequencies for the multi-sine perturbation signal is limited. The lowest (fundamental) frequency is set by the measurement time (<1 s) and the chosen number of averaging windows (two in this case) whereas the highest frequency is constrained by the gas valve, which was identified in a separate experiment: for frequencies higher than 50 Hz the valve is unable to follow the requested trace. Furthermore, earlier experiments indicated that the total expected plasma response time after a sudden increase in gas injection is in the order of 15 ms[29], additionally limiting the available frequency range.

*Number of frequencies*. Choosing fewer frequencies in the multi-sine perturbation signal results in a higher SNR per excited frequency, but less information to construct the FRF. Here, the number of applied multi-sine frequencies was chosen differently for the L-mode and H-mode experiments: The additional ELM disturbance in H-modes lowers the SNR, so fewer input frequencies, 3 instead of 4, were chosen.

*Frequency selection*. Common nonlinear system dynamics lead to clear contributions in measured system response at non-excited frequencies, particularly at integer multitudes (harmonics) of the excited frequencies[35,42]. To disentangle and quantify such effects in the measurement, some harmonics are removed in the perturbation[40]. Choosing integer harmonics of the fundamental frequency ensures that all harmonics fit exactly into the measurement window. Figure 5 summarizes the selected frequencies for the various experiments.

*Amplitude*. The amplitude of the perturbation must be sufficient to obtain a detectable response. It must, however, remain small enough for the response to be (dominantly) in the linear regime. In this particular case, further constraints require that the measurement avoid reattachment and excitations close to the X-point, where the estimation of $L_{pol}$ is known to fail[29]. The amplitude for the H-mode experiments was chosen higher due to the expected lower sensitivity to fueling.

*Phases*. The phases of the harmonics were calculated via crest factor optimization[35]. The crest factor $C$ of a perturbation signal $u(t)$ is defined as the ratio of the squared peak value of $u$ over its energy, i.e., the ∞-norm of $u$ over the 2-norm:

$$C_f = \frac{\| u \|_\infty}{\| u \|_2} \sim \frac{\text{peak value}}{\sqrt{\text{energy}}} \quad (3)$$

Changing the phases of the harmonics in $u(t)$ whilst keeping the same amplitude spectrum leads to different 'peakedness' in the perturbed signal, and hence in $C_f$.

For a given amplitude, the lowest crest factor corresponds to a maximized energy put into the system, while minimizing the displacement of (in this case) $L_{pol}$. This maximizes the SNR for a given displacement, as is desired. We apply the common approach of taking the lowest crest factor given a large number of random phases in the multi-sine with set frequencies[35].

The tailored perturbation signals and the resulting system responses are shown in Fig. 5.

## Data availability

The data that supports the findings of this study belongs to the EUROfusion consortium and is available from the corresponding author upon reasonable request.

## Code availability

The code that was used to generate figures and analyze the data belongs to the EUROfusion consortium and is available from the corresponding author upon reasonable request.

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

## Acknowledgements

The authors would like to thank the entire TCV team (see author list of[22]) for making these experiments possible and are especially grateful to Dr. Labit for his expertize on and

modifications to the small-ELM plasma scenario. Furthermore, the entire EUROfusion MST1 team (see author list of[41]) is greatly acknowledged, in particular input from Dr. Bernert. In addition, support from J.T.W. Koenders in providing the frequency response figures is gratefully received. This work was supported in part by the Swiss National Science Foundation. This work has been carried out within the framework of the EUROfusion Consortium and has received funding from the Euratom research and training programme 2014–2018 and 2019–2020 under grant agreement No 633053. The views and opinions expressed herein do not necessarily reflect those of the European Commission.

## Author contributions

T.R. wrote the real-time image processing algorithm, lead the identification and control experiments, wrote most of the paper, and prepared most figures. M.v.B. co-designed and supervised the project. A.P. wrote the hardware interface code and real-time data processing and is the responsible officer of MANTIS. C.G. designed, ran and maintained the TCV real-time control system. B.D. is co-inventor of MANTIS and the physicist in charge of installation and integration of the diagnostic. O.F. supervised, planned and coordinated the experiments. R.v.K. designed the LPM and FRF analysis, wrote the transfer function estimation routines and aided the multisine design. F.F. co-designed and implemented the TCV real-time control system and aided the feedback/sysID integration. J.L. prepared the multisine signal and tuned the feedback controller. C.T. oversaw the experiments, designed the plasma scenarios, and is the responsible officer of the TCV gas valves. J.S. gave advice on the perturbation signal design. B.L. wrote the MANTIS data extraction routines. M.K., S.H. and D.B. oversaw and planned the experiments within the EUROfusion campaign. M.d.B. initiated and supervised the project. All authors contributed to writing the paper.

## Competing interests

The authors declare no competing interests.
