## [Peer Review File · Nature Communications]

Reviewers' Comments:

Reviewer #1:

Remarks to the Author:

This paper describes an advance in tokamak divertor control. Results are interesting and clearly presented. Background information has been extended to improve accessibility to those outside the specific field, yet remains concise. The quality of writing is excellent. Design choices are explained very well. I have identified a few minor opportunities for potential improvement, which I suggest below, along with answers to some of the journal's questions for reviewers. I would recommend acceptance of this paper after my points are addressed.

Questions for reviewers (I found two somewhat similar lists and answered both, for completeness, so sorry if there is some redundancy):

- What are the major claims of the paper? Are they novel and will they be of interest to others in the community and the wider field? If the conclusions are not original, it would be helpful if you could provide relevant references.

The major claim is that a detachment control system has been demonstrated on TCV using the new MANTIS camera diagnostic system, backed by a careful control design procedure. The system and the level of detail in its design are novel and interesting.

- Is the work convincing, and if not, what further evidence would be required to strengthen the conclusions?

Yes, the results flow together and combine into a convincing narrative. Appropriate data are shown at several logical steps.

- On a more subjective note, do you feel that the paper will influence thinking in the field? Please feel free to raise any further questions and concerns about the paper.

Yes. There are noteworthy advances compared to similar work with which I'm familiar. This work demonstrates a pathway to improving tokamak divertor control.

- We would also be grateful if you could comment on the appropriateness and validity of any statistical analysis, as well the ability of a researcher to reproduce the work, given the level of detail provided.

This work doesn't involve large numbers of shots, which limits the applicability of some questions of statistics, although I can say I am convinced by the argument as to why tomographic inversion of the camera data is not required, and I am satisfied with the treatment of sources of noise, both in processing measurements and in planning. Also, the experiments have programmed a reference that changes its value in a way that I would not expect to be followed coincidentally, which is important. The steps and bumps in reference programs should be enough to rule out coincidental agreement with measurements; I am convinced that this is what things should look like if the controller is successful in influencing the measurement to be close to the reference. There is adequate information given to reproduce the system, assuming one obtains a sufficient description of the MANTIS camera from one of the references.

- What are the noteworthy results?

Feedback control of the detachment front of a tokamak using a camera system, including very formal and detailed descriptions of the design process, controller, system identification, etc, and demonstration of a high quality system implemented based on control design theory.

- Will the work be of significance to the field and related fields? How does it compare to the established literature? If the work is not original, please provide relevant references.

Detachment control in tokamaks has been described before (as referred to by the paper), but this paper is significant because it takes the subject farther in two main respects: the sensor used for the control scheme is new and more complicated and hopefully more powerful than others used so far as far as I know, and the process of system identification and control design is more formal and rigorous, leading to the controller itself seemingly working immediately as a result, in contrast to some previous reports of getting a functional controller that needs manual fine tuning to have good performance.

- Does the work support the conclusions and claims, or is additional evidence needed?

Yes, the conclusions are well supported. The authors also discuss limitations of certain specific features of their implementation and a reasonable outlook for future work. Namely, the gas species in question must change, since carbon must be avoided in future devices. This will have consequences. However, assuming a gas species with usable emission lines (that are excited in a relevant temperature range) is in the plasma, the technique should be adaptable. This issue is adequately acknowledged. As for the performance of this specific implementation, it is analyzed and presented in good detail, with satisfying explanations for all the important observations.

- Are there any flaws in the data analysis, interpretation and conclusions? - Do these prohibit publication or require revision?

No, there is no flaw that I can see in the analysis or interpretation; the treatment of noise, disturbances, and sources of error seems to be complete, the logic is all quite sound, and the conclusion is straightforward based on the evidence shown: that the controller works well in different types of TCV discharges. As mentioned before, the experimenters used changes in the reference program to avoid coincidental agreement between measurement and reference.

- Is the methodology sound? Does the work meet the expected standards in your field?

Yes. There are very thorough explanations to support key design choices. The measurements come from a diagnostic that has been described and reviewed already, although it is still pretty new.

- Is there enough detail provided in the methods for the work to be reproduced?

Yes. Key equations and hardware descriptions are provided. If I had the resources to build a MANTIS camera, and some time on a tokamak to use it, I think I could reimplement this for myself based on the descriptions here.

Suggested revisions:

1. Page 1: You have defined the divertor targets as the impact points at the ends of the divertor legs, but "divertor target" traditionally denotes an area (or line on the 2D projection), whereas the end of the divertor leg is usually called the strike point. For example, see figure 4.3 of the book "The Plasma Boundary of Magnetic Fusion Devices" by P. C. Stangeby (2000).
2. Page 2: You mention that pressure and power drop in detachment, which is described correctly, but I think you should also emphasize that this drop in pressure along the field lines is not normally present without intervention (it may seem obvious within the field, but this is not a fusion-specific journal). TCV only survives putting its full power on the tiles because it doesn't have as much power to put out as ITER/DEMO.
3. Page 4: "wall pumping" should probably be defined for a general audience. Or perhaps you might just say "... dominated by sticking to the walls" or something like that.
4. Page 8: "three proportional-integral (PI) controllers on (2)" means "... controllers on equation (2)", right? Anyway, please clarify what (2) is.

Reviewer #2:

Remarks to the Author:

In "Real-time feedback control of the impurity emission front in tokamak divertor plasmas" Dr. Ravensbergen et al. present a new combined system to control a indicator of 'plasma detachment' (not the detachment), a regime characterized by both a large reduction in plasma pressure and temperature at the divertor target, at TCV fusion reactor. This system begins three parts together:

1. A real-time camera system that look at the Carbon radiation
2. Analysis system that find the edge of rough detachment location (1D parameter) from the image.
3. PI controller to control single-input-single-output (SISO) to adjust the gas valve to keep the radiation edge at a requested location. Here is the output is the distance of the radiation edge, and the input is the gas input to the fusion tokamak.

Then this system is used at TCV to successfully control the edge of the detected system in real experiments.

The paper is well-written. All the figures and analysis is appropriate and methodology is almost spotless. The paper is ready for publication at a journal with minor edits. There are minor reference issues that needs fixing and some additional information that should be included. These are usual minor referee comments.

The main question I have as the reviewer is more on the appropriateness of this paper for nature communications (or maybe Nuclear Fusion or Fusion Engineering and Design or similar journals would be more appropriate) This needs to be decided by the editor after reading the reviewer comments below. Since Nat. Comm. is an interdisciplinary journal, it is very hard for me to judge this. I will list the items in the paper that would help make this decision:

1. The real-time camera and the analysis for finding the edge of carbon radiation threshold for rough detachment location is already published. It is cited in this paper. This is not new and does not qualify the paper of Nat. Comm. Also, some of the authors from this paper and others from the DIFFER group (the institution of many of the authors) and EAST group have already shown more sophisticated image processing that allows finding of the full 2D surface shape of the tokamak plasma. This is a harder

task than the detection of a single parameter, distance to the divertor, from a camera image. See doi: 10.1186/s40064-016-3697-9, <https://doi.org/10.1063/1.3499219> and https://www.differ.nl/news/phd_defense_gillis_hommen_optical_detection_helps_control_shape_of_fusion_plasmas

2. The paper spends a good deal of time explaining the great job of system id / control design for the PI controller. Fusion control systems are usually designed ad hoc. Though within the last decade there have been numerous fusion control design papers. However, at the end of the day Proportional-Integral control is only two paper control systems and there are commercial tools that can find these parameters automatically and control engineer without a phd should be able to tune these two parameters. This PI control design does not qualify the Nat. Com. publication.

3. Finally the PI control based on camera analysis is used to control a proxy for detachment. Control of detachment has been demonstrated many times in various machines. Using real-time Thomson Scattering, Bolometer, current tiles etc. This is not a new capability and does not qualify for Nat. Comm.

4. There is no associated physics analysis so we can not to judge the additional new physics capability that the system brings compared to other systems.

Based on the notes above it is clear that it is the unified capability that would be the reason for the acceptance at Nat. Comm. As this is not a physics paper, the engineering aspects needs to show the clear advantage over current state-of-the-art in practically. I believe, just having a nice system that is functional at a fusion test reactor that is not representative of a fusion reactor is not enough. Then, one would like to learn how the new system overcomes the issues identified in other detachment control systems. I am going to note a few short comings the detachment control schemes that maybe this system can solve (or already solved). If so, the authors should make this clear. Here are a few problems that detachment control community faces that if this system can solve, it would be helpful to include in the publication:

1. The gas pipes for ITER and future reactors are very long thus can not allow fast response. At ITER the gas time time is ~ 1 second (from the time a command goes to the time gas starts flowing and reaching the plasma). The design for reactor is not fully set yet but it is normal expect longer time delays. Pellet systems which would the main way to fuel are again very slow and more importantly they trigger ELMs which tend to move the detachment front all the way to the strike point.

Would this system allow a detachment control for say ITER/DEMO like time delays? This is a very big issue. If one uses this system with pellets which would trigger ELMs and long time delays, is it possible to control detachment front for DEMO?

2. ITER is designed to have divertor Thomson scattering system and other diagnostics that would give detachment measurement. Would this system outperform these diagnostics?

3. There is a known 'T_e cliff' phenomenon that leads to dithering (edge of the detachment goes to the strike point or the x-point, and can not be stabilized in between) in many cases does not allow some (most???) H-mode regimes to be controlled.

McLean A.G. et al 2015 Electron pressure balance in the SOL through the transition to detachment J. Nucl. Mater. 463 533, D. Eldon et al 2017 Nucl. Fusion 57 066039, Jaervinen A.E. et al 2018 E×B flux driven detachment bifurcation in the DIII-D tokamak Phys. Rev. Lett. 121 075001

Would this system allow the control of T_e cliff or avoid it?

4. Related to this, for most tokamaks, the distance from the x-point to the strike point is very short to

have more area for fusion production (fill the machine with plasma). Your system can control within ~ 5 cm range. This does not seem to be better than other systems. Is there an advantage of your system on this front? More importantly, as the leg gets shorter, the plasma tends to attach to the x-point of the strike point. This system seems to be designed for long divertor leg systems. Can this work outperform other detachment control systems for shorter leg systems?

5. Neutron damage and capability to run for diagnostics for fusion reactors is a big problem. This system is using Carbon for measurements. Though it is talked about a bit in the text, is it possible to easily build a system that works with Tungsten instead of Carbon? Does this system have an advantage vs other options (say bolometers) in reactor environment?

To sum up, the paper is ready to be published with a few minor modifications. The main thing that the Editor needs to decide is the fit to the Nat. Comm. Authors did not make it very clear how this system pushes the state-of-the-art vs the other detachment control publications and how it fits in Nat. Comm.

Other issues:

1. Real-time Thomson system gives the best detachment measurement, as it measures the electron Temperature along the divertor leg. This system is used for realtime detachment control at DIII-D (<https://doi.org/10.1016/j.jnucmat.2014.11.099>) and it is part of the ITER diagnostics set. Also, DIII-D experiments did use D for detachment controls and other impurities. Please note this in the paper clearly. You should compare the advantages of Carbon radiation vs bolometers vs Divertor Thomson etc. in the paper.

2. G.S. Xu et al 2020 Nucl. Fusion 60 086001, Divertor impurity seeding with a new feedback control scheme for maintaining good core confinement in grassy-ELM H-mode regime with tungsten monoblock divertor in EAST, is missing from references.

Reviewer #3:

Remarks to the Author:

The manuscript discusses a feedback control scheme for detached divertor plasma. Control of the detached divertor is an urgent topic in the plasma fusion devices. The manuscript shows remarkable results on the topic, thus it is reasonable to be published in Nature comm. Thus, I do generally recommend this manuscript.

Before publication, it is better to revise the following points;

The paper is mainly discussing the feedback control system. However, in the introduction part, the novelty of their control system is not well explained. Instead, the introduction discusses the novelty of diagnostic for the detachment front. It is better to emphasize the difference in the control system.

Relating to the previous point, the multi-spectral imaging of the MANTIS diagnostic was not used in this study. Only one spectrum (465nm) was used. Of course, the use of the MANTIS will be a very novel technique for other metallic wall devices, but currently, the MANTIS may not be the novelty of this study.

In figure 2, the (a) shows an entire vessel image, and the (b) shows a divertor leg space image as explained in the main text. However, the images and the caption is misleading as if both images show the entire vessel. Please correct.

The divertor port of TCV has a wide and open space and there are enough rooms for the control. Then,

it is better to discuss about the applicability of the developed scheme to other devices.

In section 2, the ability of the real-time tracking system is discussed. Please add its time-resolution and comparison with other schemes.

In section 3, "Reliable, systematic off-line feedback controller design with acceptable performance, " This sentence is too general. Please explain what is "acceptable"?

In section 3.1, "The piezo-type gas valve [36] controls ..." Do you need this sentence?

In section 3.1, "it becomes more appropriate to maximize the input energy density..." What is "input energy density"?

Do you really need section 3.2?

In figure 5, the gas valve voltage shows negative values. What does this mean? The valve is closed at $V=0$ or shutting off signal is corresponding to a negative voltage? Then, what are the ripples in the negative u_{dc} region indicating?

In figure 5, the caption says "a high noise floor at (c)". It looks like the noise floor indicated by the black dash lines are similar for (b) and (c).

In section 3.3, the end of page 7, what is the "standard gas-valve dynamic models"?

Figure 6, the legend "#62163, #65305, #65309" are corresponding to "(b), (a), (c)" of Figure 5, respectively. It is better to line up with a same order.

On page 8, " For such rational systems, each pole results in a -20 dB/decade gain change and -90 ° phase change1. " The "1" seems typo.

In figure 6, the slope of -10dB/dec should appear as discussed in the main text.

Please explain the differences between baffled and unbaffled experiments.

In section 4, is there any particular reason for doing the stair-case and the bump-case experiments? Why not showing the same cases for different modes?

In figure 7, the caption tells the results are perturbed by the ELM condition change during the discharge. Then it is better to show the temporal changes of the ELM size and frequency or the core density in this shot.

On behalf of all the authors, I kindly thank the reviewers for their helpful questions and suggestions. A detailed response to their points is given in the attached document. The original points of the reviewers are given in black, and our responses are marked in blue. Modifications in the manuscript are highlighted in red.

With kind regards,

Timo Ravensbergen

Reviewer #1 (Remarks to the Author):

This paper describes an advance in tokamak divertor control. Results are interesting and clearly presented. Background information has been extended to improve accessibility to those outside the specific field, yet remains concise. The quality of writing is excellent. Design choices are explained very well. I have identified a few minor opportunities for potential improvement, which I suggest below, along with answers to some of the journal's questions for reviewers. I would recommend acceptance of this paper after my points are addressed.

Answers of reviewer 1 to the 'Questions for Reviewers'

Questions for reviewers (I found two somewhat similar lists and answered both, for completeness, so sorry if there is some redundancy):

- What are the major claims of the paper? Are they novel and will they be of interest to others in the community and the wider field? If the conclusions are not original, it would be helpful if you could provide relevant references.

The major claim is that a detachment control system has been demonstrated on TCV using the new MANTIS camera diagnostic system, backed by a careful control design procedure. The system and the level of detail in its design are novel and interesting.

- Is the work convincing, and if not, what further evidence would be required to strengthen the conclusions?

Yes, the results flow together and combine into a convincing narrative. Appropriate data are shown at several logical steps.

- On a more subjective note, do you feel that the paper will influence thinking in the field? Please feel free to raise any further questions and concerns about the paper.

Yes. There are noteworthy advances compared to similar work with which I'm familiar. This work demonstrates a pathway to improving tokamak divertor control.

- We would also be grateful if you could comment on the appropriateness and validity of any statistical analysis, as well the ability of a researcher to reproduce the work, given the level of detail provided.

This work doesn't involve large numbers of shots, which limits the applicability of some questions of statistics, although I can say I am convinced by the argument as to why tomographic inversion of the camera data is not required, and I am satisfied with the treatment of sources of noise, both in processing measurements and in planning. Also, the experiments have programmed a reference that changes its value in a way that I would not expect to be followed coincidentally, which is important. The steps and bumps in reference programs should be enough to rule out coincidental agreement with measurements; I am convinced that this is what things should look like if the controller is successful in influencing the measurement to be close to the reference. There is adequate information given to reproduce the system, assuming one obtains a sufficient description of the MANTIS camera from one of the references.

- What are the noteworthy results?

Feedback control of the detachment front of a tokamak using a camera system, including very formal and detailed descriptions of the design process, controller, system identification, etc, and demonstration of a high quality system implemented based on control design theory.

- Will the work be of significance to the field and related fields? How does it compare to the established literature? If the work is not original, please provide relevant references.

Detachment control in tokamaks has been described before (as referred to by the paper), but this paper is significant because it takes the subject farther in two main respects: the sensor used for the control scheme is new and more complicated and hopefully more powerful than others used so far as far as I know, and the process of system identification and control design is more formal and rigorous, leading to the controller itself seemingly working immediately as a result, in contrast to some previous reports of getting a functional controller that needs manual fine tuning to have good performance.

- Does the work support the conclusions and claims, or is additional evidence needed?

Yes, the conclusions are well supported. The authors also discuss limitations of certain specific features of their implementation and a reasonable outlook for future work. Namely, the gas species in question must change, since carbon must be avoided in future devices. This will have consequences. However, assuming a gas species with usable emission lines (that are excited in a relevant temperature range) is in the plasma, the technique should be adaptable. This issue is adequately acknowledged. As for the performance of this specific implementation, it is analyzed and presented in good detail, with satisfying explanations for all the important observations.

- Are there any flaws in the data analysis, interpretation and conclusions? - Do these prohibit publication or require revision?

No, there is no flaw that I can see in the analysis or interpretation; the treatment of noise, disturbances, and sources of error seems to be complete, the logic is all quite sound, and the conclusion is straightforward based on the evidence shown: that the controller works well in different types of TCV discharges. As mentioned before, the experimenters used changes in the reference program to avoid coincidental agreement between measurement and reference.

- Is the methodology sound? Does the work meet the expected standards in your field?

Yes. There are very thorough explanations to support key design choices. The measurements come from a diagnostic that has been described and reviewed already, although it is still pretty new.

- Is there enough detail provided in the methods for the work to be reproduced?

Yes. Key equations and hardware descriptions are provided. If I had the resources to build a MANTIS camera, and some time on a tokamak to use it, I think I could reimplement this for myself based on the descriptions here.

Suggested revisions:

1. Page 1: You have defined the divertor targets as the impact points at the ends of the divertor legs, but “divertor target” traditionally denotes an area (or line on the 2D projection), whereas the end of the divertor leg is usually called the strike point. For example, see figure 4.3 of the book “The Plasma Boundary of Magnetic Fusion Devices” by P. C. Stangeby (2000).

We have modified the manuscript accordingly, which now reads: “The two branches of the X-point that do not enclose the confined region eventually lead to the machine walls and are called the divertor legs and their impact points, the divertor strike points.”

2. Page 2: You mention that pressure and power drop in detachment, which is described correctly, but I think you should also emphasize that this drop in pressure along the field lines is not normally present without intervention (it may seem obvious within the field, but this is not a fusion-specific journal). TCV only survives putting its full power on the tiles because it doesn't have as much power to put out as ITER/DEMO.

We have updated the sentence the reviewer refers to, such that it now reads: “(...)These processes [the radiating/collisional loss processes, previous sentence], when present to a sufficient degree, lead to the formation of a pressure and power drop that is not observed during usual operation and is termed detachment”

3. Page 4: “wall pumping” should probably be defined for a general audience. Or perhaps you might just say “... dominated by sticking to the walls” or something like that.

The sentence now reads “Particle removal in the divertor is dominated by absorption by the first wall material, known as wall pumping, which cannot be actively controlled.”

4. Page 8: “three proportional-integral (PI) controllers on (2)” means “... controllers on equation (2)”, right? Anyway, please clarify what (2) is.

This was indeed supposed to be a reference to equation (2). Changed in the manuscript.

Reviewer #2 (Remarks to the Author):

In "Real-time feedback control of the impurity emission front in tokamak divertor plasmas" Dr. Ravensbergen et al. present a new combined system to control a indicator of 'plasma detachment' (not the detachment), a regime characterized by both a large reduction in plasma pressure and temperature at the divertor target, at TCV fusion reactor. This system begins three parts together:

1. A real-time camera system that look at the Carbon radiation
2. Analysis system that find the edge of rough detachment location (1D parameter) from the image.
3. PI controller to control single-input-single-output (SISO) to adjust the gas valve to keep the radiation edge at a requested location. Here is the output is the distance of the radiation edge, and the input is the gas input to the fusion tokamak.

Then this system is used at TCV to successfully control the edge of the detected system in real experiments.

The paper is well-written. All the figures and analysis is appropriate and methodology is almost spotless. The paper is ready for publication at a journal with minor edits. There are minor reference issues that needs fixing and some additional information that should be included. These are usual minor reviewer comments.

The main question I have as the reviewer is more on the appropriateness of this paper for nature communications (or maybe Nuclear Fusion or Fusion Engineering and Design or similar journals would be more appropriate) This needs to be decided by the editor after reading the reviewer comments below. Since Nat. Comm. is an interdisciplinary journal, it is very hard for me to judge this. I will list the items in the paper that would help make this decision:

1. The real-time camera and the analysis for finding the edge of carbon radiation threshold for rough detachment location is already published. It is cited in this paper. This is not new and does not qualify the paper of Nat. Comm. Also, some of the authors from this paper and others from the DIFFER group (the institution of many of the authors) and EAST group have already shown more sophisticated image processing that allows finding of the full 2D surface shape of the tokamak plasma. This is a harder task than the detection of a single parameter, distance to the divertor, from a camera image. See doi: 10.1186/s40064-016-3697-9,

<https://doi.org/10.1063/1.3499219> and https://www.differ.nl/news/phd_defense_gillis_hommen_optical_detection_helps_control_shape_of_fusion_plasmas

The real-time use of cameras for detachment control is, in fact, entirely new and unprecedented in the field, and more sophisticated than earlier use of cameras in control for tokamaks. We will argue in a few points why this is the case.

1. The reconstruction of the emission front location is without a doubt more sophisticated than reconstruction of the plasma shape. Reconstructing the full shape of the divertor leg is actually the first step in pinpointing the emission front, after which additional, even non-linear and thus computationally heavier, steps are necessary to pinpoint the emission front along this leg. Our team developed a new real-time image processing algorithm specifically tailored to this challenge. Note that the original optical plasma shape reconstruction was developed by a number of our team members (<https://doi.org/10.1063/1.3499219>).
2. Our detection algorithm in itself is not the novelty of this study and is indeed already published: This is clearly stated in the manuscript with the appropriate citation, e.g., T. Ravensbergen *et al.*, *Nucl. Fus.* **60**, 2020. We would like to stress nonetheless that this paper discussing the emission front algorithm is strictly limited to post-discharge analysis and does not show actual real-time control. Such real-time control is evidently the novelty of this manuscript and the results therein.
3. While the previously published real-time optical boundary reconstruction of the core plasma for shape control indeed involves a larger portion of the plasma, both this previous work and our approach presented here are 1D reconstructions: both the core and the divertor plasma reconstruction rely on parametrized fits through the detected points in the image.
4. One of the key differences between our algorithm and the referenced algorithms by reviewer 2 is that our algorithm uses a Derivative-of-Gauss kernel in the feature detection instead of the simpler Sobel kernel. We have tested the Sobel kernel in our first version, and it failed to detect the leg properly. The reason for this is that detection of the relevant feature in the camera image (the optical plasma boundary) is easier for the core than for the divertor plasma: The 'poloidal width' of the optical boundary of the core plasma is approximately constant for the whole discharge, whereas the poloidal feature of the divertor leg is constantly changing based on the local plasma transport and temperature. This makes the relatively simple Sobel kernel for edge detection, as it was introduced by Hommen *et al.* and later used by the EAST group (eq. 4 in the Shu *et al.* paper reviewer 2 refers to) *not* applicable to our goal. Our use of Derivative-of-Gauss kernels, the resulting detection of the divertor leg, and the mapping thereof to the poloidal cross-section of the machine are not shown in this paper, but are explained in T. Ravensbergen *et al.*, *Nucl. Fus.* **60**, 2020.
5. The change in emission intensity in the divertor plasma is so significant that our algorithm alone, and certainly the algorithms referenced by reviewer 2, are not sufficient. Hence, in addition, our team made adjustments to the hardware interface of the camera-setup, such that the gain and exposure time of the camera(s) is adjusted in real-time. This makes sure that the images that are fed to the detection algorithm are not over-exposed. Such a modification is not strictly necessary for optical boundary reconstruction of the core plasma, as the intensity changes observed there are less severe. See A. Perek *et al.*, *RSI* **90** 2019 for details.

2. The paper spends a good deal of time explaining the great job of system id / control design for the PI controller. Fusion control systems are usually designed ad hoc. Though within the last decade there have been numerous fusion control design papers. However, at the end of the day Proportional-Integral control is only two paper control systems and there are commercial tools that can find these parameters automatically and control engineer without a phd should be able to tune these two parameters. This PI control design does not qualify the Nat. Com. publication.

The design and choice of the PI controller is not the reason for submission to Nat. Comm. and we fully agree with reviewer 2 that this is not the novelty of our work. What is novel, however, is the system identification approach. In our view, this system identification approach is one of the key contributions of our work to the fusion community, especially the plasma exhaust. This believe is also our motivation to discuss the system identification procedure in such great detail. System identification is important in the sense that it allows rigorous and formal comparison of detachment dynamics across regimes and devices. In the paper, we show that system identification is challenging to characterize the divertor plasma due to the many disturbances and short measurement time. Our perturbation signal design and analysis method are unprecedented in their ability to extract reliable data from the perturbative experiments, which can then be used for controller design.

As the reviewer rightly points out, in the end, the PI controller is simple and only contains two parameters. Our choice for a PI controller is justified by an emphasis on robustness over performance given the novelty of our experimental setup. However, we would like to stress that tuning such a controller always requires dynamic understanding, even with automatic approaches, and often in the form of a frequency response function (FRF). We obtained such an FRF using state-of-the-art system identification methods. In this application domain,

obtaining a sufficiently high SNR to get a reliable FRF is a challenge, for which automated methods are not (yet) equipped.

3. Finally the PI control based on camera analysis is used to control a proxy for detachment. Control of detachment has been demonstrated many times in various machines. Using real-time Thomson Scattering, Bolometer, current tiles etc. This is not a new capability and does not qualify for Nat. Comm.

The novelties in our work are the use of cameras in the divertor and the unified *systematic approach* towards detachment control, especially the methodology to obtain a reliable Frequency Response Function for controller design. The advantage of using cameras over other diagnostics will be discussed in detail at point 2 (below) of the reviewers remarks regarding diagnostics for detachment control in future machines below. In the manuscript, we have added a paragraph in the discussion section providing a direct comparison between our control approach using cameras and other existing approaches.

4. There is no associated physics analysis so we can not judge the additional new physics capability that the system brings compared to other systems.

The real-time feedback loop including MANTIS and the gas valve does not yield new physics insight in itself. However, the dynamic measurements of the emission front response upon fueling are an important physics result which is not published elsewhere. In the future, the obtained FRFs will be compared to similar measurements on other machines and across regimes to better understand the scaling of the detachment dynamics. This is especially relevant for larger, high power machines in which numerous challenges regarding detachment control are foreseen. Some of these challenges are in fact pointed out by the reviewer below. It is clear in our manuscript that our focus is not physics studies with MANTIS. However, MANTIS is well equipped for such studies as it was designed with both control and post-shot analysis in mind. See A. Perek *et al.*, *RSI* **90** 2019 for details.

Based on the notes above it is clear that it is the unified capability that would be the reason for the acceptance at Nat. Comm. As this is not a physics paper, the engineering aspects needs to show the clear advantage over current state-of-the-art in practically. I believe, just having a nice system that is functional at a fusion test reactor that is not representative of a fusion reactor is not enough. Then, one would like to learn how the new system overcomes the issues identified in other detachment control systems. I am going to note a few short comings the detachment control schemes that maybe this system can solve (or already solved). If so, the authors should make this clear. Here are a few problems that detachment control community faces that if this system can solve, it would be helpful to include in the publication:

1. The gas pipes for ITER and future reactors are very long thus can not allow fast response. At ITER the gas time time is ~ 1 second (from the time a command goes to the time gas starts flowing and reaching the plasma). The design for reactor is not fully set yet but it is normal expect longer time delays. Pellet systems which would be the main way to fuel are again very slow and more importantly they trigger ELMs which tend to move the detachment front all the way to the strike point.

Would this system allow a detachment control for say ITER/DEMO like time delays? This is a very big issue. If one uses this system with pellets which would trigger ELMs and long time delays, is it possible to control detachment front for DEMO?

We do not know whether our control (or any other detachment control currently performed) would be able to deal with the relevant time scales of a next generation device, mainly because those time scales are unknown. Our work, however, notably the system identification method, provides a means to systematically characterize these time scales in machines in operation today which could assist in the scaling towards larger machines. It is therefore the *approach* presented in the manuscript that will help study detachment control for future devices, regardless the specific choice of actuator and diagnostic.

Furthermore, in future machines, controller design based on models is key, as the experimental time for controller tuning is limited. In previous work (Ravensbergen *et al.*, *Nucl. Fusion* **58** (2018) 016048) we have applied non-causal and model-based control techniques to deal with large time delays in ITER's density control. Our integrated approach to control presented here, particularly the system identification, is key in studying such methods and reducing experimental time.

One of the impacts of our work is that the EU DEMO design now considers spectroscopy with a tangential view (W. Gonzalez *et al.*, 2020 *JINST* **15** C01008). This setup is similar to ours in terms of what it measures (spectral lines) and its view. However, detachment control using this diagnostic still needs to be largely developed, also on

machines in operation today. We are convinced that MANTIS and our system identification/control approach will play an important part in this development.

2. ITER is designed to have divertor Thomson scattering system and other diagnostics that would give detachment measurement. Would this system outperform these diagnostics?

It is not our goal to outperform other real-time capable detachment diagnostics, but to provide a meaningful diagnostic addition and to highlight the benefits of a systematic approach. Nevertheless, the MANTIS diagnostic is indeed advantageous on multiple aspects with respect to other divertor plasma diagnostics, which we will summarize below. Whether these points will lead to MANTIS outperforming other diagnostics on a future machine is unknown, and significantly depends on the detachment characteristics in ITER.

1. MANTIS is a 2D resolved imaging diagnostic, which makes it compatible with changing magnetic configurations, for example during strike point sweeping. This is a clear advantage over Thomson scattering, which entails a series of measurements in fixed and notably small plasma volumes. It is important to note that these points do not necessarily align with the (full) divertor leg.
2. The acquisition frequency of MANTIS is 200-1000 Hz, making it generally faster than Thomson scattering (50-80 Hz), but slower than interferometry or probes (>1 kHz). Higher acquisition frequencies are important, as they generally lead to better SNR in the FRF estimation, as is discussed in the manuscript.
3. MANTIS' line of sight is tangential to the plasma, such that (real-time approximations of) inversions are not necessary for control purposes.
4. A full 2D view of the poloidal divertor plasma is obtained using MANTIS. This is particularly relevant, as it has become clear that the onset of detachment is driven from a high density, low temperature region in the private flux region (Lipschultz *et al.*, Phys. Plasmas **6** 1999), which makes analysis of diagnostics with a line-of-sight across the divertor leg more challenging.
5. MANTIS is equipped with multiple spectral filters which can be used to obtain the spatial distribution of the volumetric processes associated with detachment, and, as such, can be used to control not just emission fronts but also recombination and/or ionization fronts in real-time by approximating those fronts using Balmer line ratios.
6. Feedback control using MANTIS is possible in attached, marginally/partially detached, and (strongly) detached conditions. This is not the case for probes (only attached or marginally detached), and for example AXUV photodiodes (only detached). See (Xu *et al.* Nucl. Fusion **60** 2020) for details.

3. There is a known 'T_e cliff' phenomenon that leads to dithering (edge of the detachment goes to the strike point or the x-point, and can not be stabilized in between) in many cases does not allow some (most???) H-mode regimes to be controlled.

McLean A.G. et al 2015 Electron pressure balance in the SOL through the transition to detachment J. Nucl. Mater. 463 533, D. Eldon et al 2017 Nucl. Fusion 57 066039, Jaervinen A.E. et al 2018 ExB flux driven detachment bifurcation in the DIII-D tokamak Phys. Rev. Lett. 121 075001

Would this system allow the control of T_e cliff or avoid it?

Irrespective of the diagnostic, the bifurcation-like jump from attached to detached conditions that is described in the paper is indeed too fast for feedback control relying on gas. However, our system identification approach will in fact help to systematically assess the effect of gas on the dithering state. Furthermore, the promising regime known as the x-point radiator (Bernert *et al.* NF submitted) can be analyzed and controlled using our approach and MANTIS.

4. Related to this, for most tokamaks, the distance from the x-point to the strike point is very short to have more area for fusion production (fill the machine with plasma). Your system can control within ~5 cm range. This does not seem to be better than other systems. Is there an advantage of your system on this front? More importantly, as the leg gets shorter, the plasma tends to attach to the x-point of the strike point. This system seems to be designed for long divertor leg systems. Can this work outperform other detachment control systems for shorter leg systems?

Our approach is not necessarily restricted to short leg systems. In fact, on the WEST tokamak (a machine with a particularly short divertor leg) emission is seen to move away from the target upon detaching, allowing a detectable response.

5. Neutron damage and capability to run for diagnostics for fusion reactors is a big problem. This system is using Carbon for measurements. Though it is talked about a bit in the text, is it possible to easily build a system that

works with Tungsten instead of Carbon? Does this system have an advantage vs other options (say bolometers) in reactor environment?

The reviewer rightly raises two issues. The first is associated with spurious light sources: An issue that is highly relevant for most camera systems is reflection of light on the first wall. This is particularly problematic in tokamaks with metallic plasma facing components. However, as we employ a tangential view, we expect that the separation between plasma emission and reflected light can be done using simple polarizers.

The second issue is associated with the location of radiation of the impurities. No detachment control system is or will be designed to work with tungsten, as intrinsic impurity species from the wall (Be or W) radiate in the core plasma at temperatures generally much higher than those typically associated with the SOL. Machines equipped with a metallic first wall therefore rely on extrinsic impurity seeding to induce radiation in the divertor. As such, MANTIS and our approach in general are not meant to study tungsten emission. When seeding, impurity emission fronts arise in the charge states associated with the seeding species, for example N-II when seeding nitrogen gas. These fronts have similar features in the image, such that reliable detection is straightforward. Without precaution all diagnostics relying on a direct line-of-sight into the plasma suffer from neutron damage. Recent preliminary assessment, however, has indicated that a direct line of sight to the divertor plasma is possible for imaging systems even in a DEMO-like neutron environment (Biel *et al.*, *FED*, **146**, 2019). The currently foreseen detachment control diagnostic is therefore a spectrometer with a tangential view (W. Gonzalez *et al.*, 2020 JINST **15** C01008), yielding a setup that is particularly similar to ours, as explained in our response at point 1 above.

To sum up, the paper is ready to published with a few minor modifications. The main thing that the Editor needs to decide is the fit to the Nat. Comm. Authors did not make it very clear how this system pushes the state-of-the-art vs the other detachment control publications and how it fits in Nat. Comm.

We have summarized the advantages of a systematic approach as we present in the paper in the responses above, and we have tried to modify the manuscript in such a way that it reflects these points better.

Other issues:

1. Real-time Thomson system gives the best detachment measurement, as it measures the electron Temperature along the divertor leg. This system is used for realtime detachment control at DIII-D (<https://doi.org/10.1016/j.jnuclmat.2014.11.099>) and it is part of the ITER diagnostics set. Also, DIII-D experiments did use D for detachment controls and other impurities. Please note this in the paper clearly. You should compare the advantages of Carbon radiation vs bolometers vs Divertor Thomson etc. in the paper.

“Real-time Thomson system gives the best detachment measurement” is a very blunt statement and we have to respectfully disagree, which we will explain in detail below.

The reviewer’s main argument in favor of Real-time Thomson is that it measures the temperature along the divertor leg. We note that the reviewer on one hand insists on relevance for future devices, but on the other hand favors set-ups that are not reactor relevant. We will show in the following why a real-time Thomson scattering system is not easily applicable in a reactor environment because of three reasons: the technical implementation, the measurement, and the requirements of a future detachment control system in a reactor.

From a technical point of view, implementation of a Thomson scattering system in a reactor is not easier than a camera system. Even for ITER, the first mirrors of the Thomson scattering in the divertor will be subject to large high neutron fluxes in its nuclear campaign. This is already considered to be a challenge (E.E. Mukhin *et al.* 2014 Nucl. Fusion 54 043007), with DEMO fluxes significantly exceeding the ones in ITER.

From a physics point of view, a Thomson scattering system measures the temperature and density of the electron population. During detachment, however, the electron temperature is at best an indicator for the detached state but is not representative of the radiative loss processes taking place. Hence, its use as detachment measurement is very limited.

In this paper, we use the multispectral diagnostic MANTIS in its most basic set-up, imaging a single CIII line. We show that we can use MANTIS for real-time control with carbon as impurity species of interest. However, MANTIS is equipped with 10 cameras, and we are currently preparing the next steps to exploit this capability to further. By using multiple impurity lines for the analysis, our instrument effectively becomes a 2D thermometer. The thermal resolution is coarse (the precise temperature of every measured line depends on the plasma transport in the divertor) but the spatio-temporal resolution is superior to any TS system. Additionally, the field of view covers a significant portion of the divertor.

Finally, we stress that MANTIS has capabilities that go well beyond any RT temperature diagnostic and we are only starting to exploit its full capability. For example, recent results show that it is possible to pinpoint the 2D regions of recombination and ionization (Perek *et al.* EPS Conference on plasma physics, Milan 2019).

The relevance of Thomson scattering for detachment control is hemmed by the fact that a 2D view is essential in cases where the magnetic configuration needs to be varied. A particularly relevant example is the loss-of-detachment scenario. The response to such an event in the European DEMO baseline design is strike-point sweeping. Such sweeping aims to distribute the large heat loads during reattachment over a larger area, during which detachment control must be quickly recovered. During the sweeping of the divertor leg, the Thomson scattering alignment with the divertor leg is lost, along with its real-time data interpretation. This makes feedback in such critical conditions complicated. As our system covers the whole divertor region and resolves various lines, it is intrinsically suited for MIMO control that combines actuation on the magnetic configuration, the fuel and impurity content.

A further discussion on MANTIS can be found in the answer to the previous point 2 or reviewer 2, which includes a comparison with Thomson scattering, including its sample rate.

We have added the reviewer's suggestion to the publication regarding detachment control on DIII-D which was already in the manuscript (Eldon *et al.* *NME* **18** 2019), but which relied on bolometry. We have added comparison between the various diagnostics to the discussion section.

2. G.S. Xu et al 2020 Nucl. Fusion 60 086001, Divertor impurity seeding with a new feedback control scheme for maintaining good core confinement in grassy-ELM H-mode regime with tungsten monoblock divertor in EAST, is missing from references.

We have read this recent publication with great interest and have added it to the discussion on existing controller implementations in the introduction and discussion sections.

The updated section in the introduction now reads:

“Various feedback control solutions involving detachment have been tested experimentally, using controlled variables based on either local target measurements or from spatially resolved diagnostics. Implementations based on target measurements rely on, for example, tile current (Kallenbach *et al.*, 2012) or target ion saturation current measurements (Guillemaut *et al.*, 2017), or target proximity thermocouples (Brunner *et al.*, 2016). Among the spatially resolved methods are those based on Thomson scattering (Kolemen *et al.*, 2015) and radiated power from bolometry (Kallenbach *et al.*, 2015 and Eldon *et al.*, 2019) or AXUV (Bernert *et al.*, 2020 and Xu *et al.*, 2020). These experiments rely on (impurity) gas fueling through controlled valves as actuators. In addition, supersonic molecular beam injection (SMBI) was successfully applied as feedback actuator (Wu *et al.*, 2018).”

Reviewer #3 (Remarks to the Author):

The manuscript discusses a feedback control scheme for detached divertor plasma. control of the detached divertor is an urgent topic in the plasma fusion devices. The manuscript shows remarkable results on the topic, thus it is reasonable to be published in Nature comm. Thus, I do generally recommend this manuscript.

Before publication, it is better to revise the following points;

The paper is mainly discussing the feedback control system. However, in the introduction part, the novelty of their control system is not well explained. Instead, the introduction discusses the novelty of diagnostic for the detachment front. It is better to emphasize the difference in the control system.

As reviewer 3 rightly points out, the introduction section distinguishes existing detachment control implementations based on their choice of real-time diagnostic and actuator. This is because in our opinion, getting a proxy for the state of detachment and a means to affect its onset fit into the introduction of detachment itself. The differences in the finalized control scheme require a discussion on for example controller tuning and system identification methods. We have therefore added a paragraph in the discussion section in which the differences in the various control loops (and the steps towards a working controller) are discussed in more detail.

Relating to the previous point, the multi-spectral imaging of the MANTIS diagnostic was not used in this study. Only one spectrum (465nm) was used. Of course, the use of the MANTIS will be a very novel technique for other metallic wall devices, but currently, the MANTIS may not be the novelty of this study.

Indeed the novelty of this study is not MANTIS in itself, but its use in real-time vision-in-the-loop control for the divertor plasma.

In figure 2, the (a) shows an entire vessel image, and the (b) shows a divertor leg space image as explained in the main text. However, the images and the caption is misleading as if both images show the entire vessel. Please correct.

We added the sentence "In this image, the carbon plasma facing components on the TCV floor and inner wall are visible, but the field-of-view is restricted from the top by the outer baffle." To the main text. In the caption an additional "The field-of-view is restricted by the TCV floor (bottom) and outer baffle structure (top)" was added.

The divertor port of TCV has a wide and open space and there are enough rooms for the control. Then, it is better to discuss about the applicability of the developed scheme to other devices.

Our method of detection with the real-time algorithm and MANTIS is applicable also to shorter divertor leg systems, as the accuracy is sufficiently high. We have added a statement on the full evolution of detachment in which our method can be used to the discussion section. Whether detachment in such short legged divertors can be controlled remains to be determined,

In section 2, the ability of the real-time tracking system is discussed. Please add its time-resolution and comparison with other schemes.

A time resolution comparison between other spatial methods (TS, Bolometry, AXUV) has been added to the discussion section.

In section 3, "Reliable, systematic off-line feedback controller design with acceptable performance," This sentence is too general. Please explain what is "acceptable"?

This vague statement has been removed and we added the following sentence: "The resulting FRF can then be used for offline assessment of performance limitations and stability margins (Skogestad 2005)"

In section 3.1, "The piezo-type gas valve [36] controls ..." Do you need this sentence?

In our system identification approach, we directly identify the relationship between a gas request and the resulting position of the emission front. This sentence is therefore important, as the 'internal' dynamics of the gas valve are included in the total dynamic FRF and important in the controller design. For clarity, we have modified the sentence to "The piezo-type gas valve (...) has its own internal feedback controller aiming to track a gas puff request"

In section 3.1, "it becomes more appropriate to maximize the input energy density..." What is "input energy density"?

The energy that is contained in the input perturbation signal in the frequency range of interest. We have added an explicit "(...) input energy density of the perturbation signal" to clarify which energy we mean.

Do you really need section 3.2?

The design of specialized perturbation signals to obtain reliable measurements is a key element to our approach. Without such a signal, the SNR is simply too low to extract any meaningful transfer function necessary to develop a controller. Hence, we would like to retain section 3.2 in the manuscript.

In figure 5, the gas valve voltage shows negative values. What does this mean? The valve is closed at V=0 or shutting off signal is corresponding to a negative voltage? Then, what are the ripples in the negative u_dc region indicating?

In figure 5, the caption says "a high noise floor at (c)". It looks like the noise floor indicated by the black dash lines are similar for (b) and (c).

The time trace in Fig. 5 shows the zero-mean perturbation about the feedforward gas trace that the operator programs. In theory, this feedforward results in constant density without the perturbation. We have clarified this in section 3.3 by adding "In the experiment, this zero-mean perturbation is summed with a feedforward gas trace that keeps the core density approximately constant."

There are two effects that affect the noise floor in these figures: the quality of the measurement (particularly the MANTIS frame rate) and the disturbances during the experiment. In the (older) unbaffled experiments, the SNR is higher due to the lower MANTIS frame rate. We have rewritten the caption of Fig. 5 and have moved the

sentence in that was supposed to explain this to right after the statement on the noise floor in c).

In section 3.3, the end of page 7, what is the "standard gas-valve dynamic models"?

The models with a rational transfer function in the next sentence. This was indeed not clear. We have added an explicit "Such models" to clarify that the "rational models" are the "standard dynamic models".

Figure 6, the legend "#62163, #65305, #65309" are corresponding to "(b), (a), (c)" of Figure 5, respectively. It is better to line up with a same order.

Thanks for this suggestion, we have updated the order in Figure 5 accordingly.

On page 8, " For such rational systems, each pole results in a -20 dB/decade gain change and -90 ° phase change¹. " The "1" seems typo.

That "1" was meant to be a footnote clarifying the statement in the main text only holds for a particular class of systems. This will most likely be clearer in the final journal formatting.

In figure 6, the slope of -10dB/dec should appear as discussed in the main text.

We have added the slope to the Figure in question.

Please explain the differences between baffled and unbaffled experiments.

The purpose of baffles is explained in the beginning of Sec. 2. We have modified this particular section, which now reads: "Optional physical *baffling* in the divertor increases neutral particle compression, facilitating detachment (Reimerdes 2017). Such baffles were installed during a majority of the experiments presented in this paper." Furthermore, we have added a sentence in Sec. 3.3: "The differences in dynamic response between experiments are minor and appear to be mostly in the steady-state gain of the FRF". This is one of the outcomes of our analysis and will be in detail discussed in a future publication.

In section 4, is there any particular reason for doing the stair-case and the bump-case experiments? Why not showing the same cases for different modes?

Yes as reviewer 1 points out, requesting different reference traces for the various feedback experiments helps in arguing that the movement of the emission front is due to our controller or applied perturbation, and is not the result of the natural evolution of the discharge.

In figure 7, the caption tells the results are perturbed by the ELM condition change during the discharge. Then it is better to show the temporal changes of the ELM size and frequency or the core density in this shot.

We have added a figure to the manuscript as requested containing the density evolution for the discharge under consideration and the Da-filtered photodiode signal.

Reviewers' Comments:

Reviewer #2:

Remarks to the Author:

Authors responded to the comments and revised the paper accordingly. The paper is ready for publication.

Reviewer #3:

Remarks to the Author:

The authors revised the manuscripts well. However, a question was unanswered.

"In figure 5, the gas valve voltage shows negative values. What does this mean? The valve is closed at $V=0$ or shutting off signal is corresponding to a negative voltage? Then, what are the ripples in the negative u_{dc} region indicating? "

Once this question is answered, the paper should be published.

Reviewer #2 (Remarks to the Author):

Authors responded to the comments and revised the paper accordingly. The paper is ready for publication.

Reviewer #3 (Remarks to the Author):

The authors revised the manuscripts well. However, a question was unanswered.

"In figure 5, the gas valve voltage shows negative values. What does this mean? The valve is closed at $V=0$ or shutting off signal is corresponding to a negative voltage? Then, what are the ripples in the negative u_{dc} region indicating? "

Once this question is answered, the paper should be published.

The total gas command that is sent to the gas valve is the sum of an operator feedforward waveform and our applied perturbation, leading to a signal that is always positive. Fig. 5 only shows the applied perturbation which can be negative as long as the total gas command never becomes negative. The latter is guaranteed the case in these experiments, both by carefully checking the feedforward plus perturbation signal (which is possible since both are known before the experiment), and by a hard limit on the valve in a higher level of the control system. The goal of the operator feedforward is to keep the core density approximately constant such that the emission front position is not drifting to severely over the course of the experiment. Our applied perturbation then results in excursions around this position, such that a locally valid dynamic relationship can be extracted. The same operator feedforward is also applied during the feedback experiments. Note that in Figures 7 and 8 in the manuscript, that summarize these experiments, the operator feedforward is plotted in yellow.

In the '*Perturbative measurements and transfer function estimation*' subsection preceding the system identification experimental results, we have modified the text to further clarify our approach:

"In the experiment, this zero-mean perturbation is summed with a feedforward gas trace that keeps the core density approximately constant. The total signal that is sent to the gas valve in these experiments is therefore always positive."

In the caption of Figure 5, we have added an explicit statement that the gas valve signal is never negative:

"In the experiment, the perturbation is summed with a feedforward waveform to yield the total (always positive) gas valve command."

In text preceding the feedback control experimental results, we have clarified that the feedforward waveform is again applied:

"[Core density feedback controller is frozen when our feedback control becomes active] (...) The total gas valve command is therefore the sum of the pre-programmed feedforward waveform and the feedback control using the MANTIS-computed L_{pol} "

Reviewers' Comments:

Reviewer #3:

Remarks to the Author:

The manuscript is well revised. It is now ready for publication.